# The environmental impact of data-driven precision medicine initiatives

Gabrielle Samuel[1,2] and Anneke M. Lucassen[2,3]

[1]Department of Global Health and Social Medicine, King's College London, London, UK; [2]Wellcome Centre for Human Genetics, Oxford University, Oxford, UK and [3]Clinical Law and Ethics at Southampton (CELS), NIHR Southampton Biomedical Research Centre, University of Southampton, Southampton, UK

data-driven; digital health; precision medicine; research; environmental impacts; sustainability; e-waste

**Corresponding author:**
Gabrielle Samuel,
E-mail: gabbysamuel@gmail.com

## Abstract

Opportunities offered by precision medicine have long been promised in the medical and health literature. However, precision medicine – and the methodologies and approaches it relies on – also has adverse environmental impacts. As research into precision medicine continues to expand, there is a compelling need to consider these environmental impacts and develop means to mitigate them. In this article, we review the adverse environmental impacts associated with precision medicine, with a particular focus on those associated with its underlying need for data-intensive approaches. We illustrate the importance of considering the environmental impacts of precision medicine and describe the adverse health outcomes that are associated with climate change. We follow this with a description of how these environmental impacts are being addressed in both the health and data-driven technology sector. We then describe the (scant) literature on environmental impacts associated with data-driven precision medicine specifically. We finish by highlighting various environmental considerations that precision medicine researchers, and the field more broadly, should take into account.

## Impact statement

Precision medicine advances have driven many welcome insights into mechanisms of disease and means to treat them. However, these advances have an environmental footprint which in turn can lead to adverse health impacts. Our article argues that precision medicine research should have an interest in this environmental footprint not only because of international priority setting, but also because of its commitment to health. We describe these impacts, focusing in particular on data intensive approaches such as those associated with the energy required to collect, store, process and analyse data, as well as the materials associated with the manufacture of digital technologies, and the waste produced from them. We point to the scant discussions of the impacts of data intensive approaches to date in the health research literature, despite a growing awareness of the importance of the need for environmental sustainability within healthcare. We highlight how the carbon footprint of certain data intensive approaches compares to – for example, airline travel-and then point to various ways in which precision medicine researchers can consider the adverse environmental and health impacts of their work. Relatively simple interventions such as considerations around where, how, and when data is stored, processed, and analysed can make a significant impact on the environmental footprint of these activities. We hope our article will be of interest to a wide range of experts involved in precision medicine including policy makers.



## Introduction

Technological advances in our ability to create, link and store data relating to health have brought promises and aspirations of personalising healthcare decisions for a given patient, so that they can receive the most targeted, and therefore effective treatment (Ginsburg and Phillips, 2018). All sorts of clinical-related data – including genomic, proteomic, other 'omic' and biochemical analyses – can be linked with environmental exposure data, longitudinal information from 'wearables' and other patient-reported data, with the aim of improving care, reducing the need for unnecessary investigations and targeting therapies more appropriately. Examples that have already entered routine clinical practice are many and varied and include the treatment of certain cancers, rare genetic conditions (e.g., cystic fibrosis and Duchenne muscular dystrophy), infectious diseases (e.g., HIV) and drug responses (e.g., warfarin and codeine sensitivity) (Ashley, 2016; Ginsburg and Phillips, 2018).

These advances inevitably have a significant environmental footprint, which is sometimes justified by using consequential narratives of being necessary to improve healthcare and/or that health has intrinsic value so has a 'free pass' to not consider these issues (Samuel et al., 2022).

Nevertheless, the adverse environmental impacts of data-driven precision medicine include its effect on the climate, material environment, water and air pollution and (toxic) waste production and thus question whether this 'free pass' is appropriate. This article reviews the current literature associated with the environmental impacts of precision medicine, particularly focussing on the underlying data-intensive approach. It highlights that while much concern has focused on the environmental impacts of medicine more generally, less attention has been paid to the data-driven aspects of (precision) medicine. The review considers (1) the literature on environmental impacts studied in medicine, (2) the literature on environmental impacts associated with data-driven technologies and (3) the [scant] literature on environmental impacts of data-driven precision medicine. It concludes by highlighting various environmental considerations that precision medicine researchers, and the field more broadly, should take into account.

## Climate change and the need to consider the environmental impacts of (precision) medicine

Calls for an environmentally sustainable medicine have been made for several decades (Pierce and Jameton, 2004, Dwyer, 2009, Brown et al., 2012, Eckelman and Sherman, 2018, Richie, 2019, Health Care Without Harm, 2021). Healthcare contributes to between 1% and 5% of various global environmental impacts, including greenhouse gas emissions, particulate matter, air pollutants, reactive nitrogen in water and water use (Lenzen et al., 2020). Healthcare is also a massive emitter of waste, much of which is plastic, with single-use plastic items (syringes, blood bags and tubing) saturating everyday medical practice across the globe (Hodges, 2017). The recent COVID-19 pandemic has exemplified the issue with the generation of eight million tonnes of pandemic-associated plastic waste, primarily from hospitals (Peng et al., 2021).

While the consideration of all types of environmental impacts is important, the recent categorisation of the climate emergency (Pidgeon, 2021) has driven particular and urgent attention to environmental impacts, such as carbon dioxide and greenhouse gas emissions, that contribute to climate change. There is now no doubt that climate change is caused by human factors, resulting in an increased frequency and severity of extreme temperatures, flooding, cyclones, droughts and fire weather (IPCC, 2022). Healthcare industries contribute approximately 5.5% of a country's total emissions (as of 2014; e.g., the United States, the Netherlands, Belgium and Japan all emit approximately 8% of the country's total emissions; it is 3.3% for Mexico and 6% for Great Britain[1]) (Pichler et al., 2019). Without measures to tackle these consequences, we will see the extinction of species on land and in the ocean, as well as the devastation of environments (IPCC, 2022). Climate change also directly affects the social and environmental determinants of health – clean air, safe drinking water, sufficient food, water and secure shelter (cities, settlements and infrastructure) (World Health Organisation, 2021; IPCC, 2022). Climate events (heat, floods and cyclones) have affected food production and nutrition levels (what can be grown and the time that land can be farmed) – particularly in Africa and Central and South America (Romanello et al., 2021; IPCC, 2022). Increased exposure to extreme heat, wildfire smoke, atmospheric dust and aeroallergens have been associated with climate-related cardiovascular and respiratory distress resulting in increased morbidity and mortality (Romanello et al., 2021, IPCC, 2022). High temperatures can also reduce the frequency, duration and motivation to be physically active, in turn,

a known factor in the risk of cardiovascular disease, diabetes, cancer, cognitive decline and all-cause mortality (Romanello et al., 2021). Effects on mental health have also been documented from loss of livelihoods and culture through climate events (Romanello et al., 2021). The occurrence of malaria, dengue fever and Zika are all on the rise because increasing climate temperatures mean the geographical area where mosquitos can survive is extended, as well as their annual season, resulting in greater disease transmission (Romanello et al., 2021; IPCC, 2022). The World Health Organisation (WHO) predicts that between 2030 and 2050, climate change will cause approximately 250,000 additional deaths per year from malnutrition, malaria, diarrhoea and heat stress (World Health Organisation, 2021). While international efforts aim to limit global warming to 1.5°C, evidence shows that we must be prepared for warming up to 4°C (UK Government, 2022). In the UK, for example, the surface temperature has already risen by 1.2°C since pre-industrial times (UK Government, 2022).

## *Reducing carbon emissions and other environmental impacts of (precision) medicine*

A group of 60 countries has already committed to developing climate-resilient and/or low-carbon health systems, and nine countries have now pledged to make their healthcare systems net-zero by 2040[2] (Indonesia, Malawi, Sierra Leonne, Kenya, Liberia, Ivory Coast, Burkina Faso, Nigeria and UK). Various hospitals internationally have also signed up for the United Nation-backed 'Race to Zero' initiative.[3] For example, in England, since 2007, the National Health Service (NHS) has reduced the carbon footprint of health and social care by 18.5% (equating to the annual emissions from a small country such as Cyprus). Carbon and energy reduction initiatives have focused on a number of levels including buildings, estates and facilities; medical infrastructure, including disposable containers (McPherson et al., 2019); travel; and electronic devices such as freezers, lights and computers (NHS England, 2018).

Examples of areas in which the environmental impacts of specific health procedures and devices can be reduced include: imaging (Alshqaqeeq et al., 2020), anaesthetics (Ryan and Nielsen, 2010), inhalers (Wilkinson et al., 2019), dialysis (Moura-Neto et al., 2019), eye care (Buchan et al., 2022) and surgery (Namburar et al., 2018; Thiel et al., 2018), all of which can have particularly high environmental impacts.

Perhaps more pertinent to precision medicine research and manufacturing, global healthcare and technology companies are similarly decarbonising their biomedical research, as well as their manufacture of medical devices and pharmaceuticals (Pierce and Jameton, 2004; Hawkes, 2012; NHS England, 2018; Kmietowicz, 2021).[4] This is important, since a recent analysis has identified the pharmaceutical industry to be significantly more emission-intensive than the automotive industry (Belkhir and Elmeligi, 2019). Emissions are related to upstream manufacturing and research transportation costs for drug distribution, as well as downstream prescribing (Richie, 2021).

Developing environmentally sustainable healthcare means going beyond climate considerations to ensure natural resources are not harvested faster than they can be regenerated, or emitting waste faster than what can be assimilated by the environment (Mensah, 2019). The effects on biodiversity must be considered (Bull et al., 2022), as must water consumption. For example, health services in various countries are reducing their water consumption,[5] for example, NHS England has reduced its water footprint by 21% since 2010 – the same water volume as 243,000

Olympic swimming pools (NHS England, 2018). Precision medicine research must also attend to waste from its research laboratories. Various international initiatives have encouraged laboratories to reduce consumption, and reuse and recycle materials[6] (e.g., see Rae et al., 2022 who reviewed the environmental sustainability of neuroscience research).

## Environmental impacts of data-driven technologies

One aspect of precision medicine is data-driven research. Data-driven initiatives have resulted in an exponential increase in computing storage and processing power that has allowed precision medicine researchers to collect and collate myriad types of health-related data sets for analysis. Data-linkage studies, using a range of complex algorithms, such as machine learning and other artificial intelligence technologies are driving this 'datafication' (Ruckenstein and Schüll, 2017) of health (Erikainen and Chan, 2019), making it the fastest growing sector in the datasphere (Reinsel et al., 2018). Proteomics, metabolomics and genomics are all data-intensive solutions used by precision medicine researchers. Hogan (2020) has emphasised that by 2025, it is predicted that between 100 million and 2 billion human genomes will have been sequenced globally, using some 40 exabytes of data (Hogan, 2020). The UK 100,000 genomes project uses 21 petabytes of storage, which is equivalent storage to some 40,000 years of playback on an MP3 player (Davies, 2017). By 2025, the UK Biobank database – a leading international biobank – is expected to grow to 15 petabytes – an amount of data equivalent to that created annually by the Large Hadron Collider, though likely will be much greater than this as it continues to analyse the data it has already collected, as well as collect new data from various imaging studies.[7] Furthermore, precision medicine research of electronic health records by ML/AI techniques uses petabytes of storage (Nelson and Staggers, 2018).[8]

Whilst the hypothesised and proven benefits are varied, the adverse consequences of precision medicine's environmental footprint require recognition and consideration. Digital and data-driven technologies are often described using metaphors of immateriality (connecting 'virtually') or fluffiness and transparency (computing in a 'cloud'), yet their physical presence is real, comprised of a multitude of computers, servers, cables and wires (Holt and Vonderau, 2015; Lucivero, 2020). Large and expansive data centres house data servers, and physical digital infrastructures supply information and communication technologies (ICTs). And, while data centres are often portrayed in environmentally friendly ways (e.g., surrounding trees, images of clean and shiny servers) (Holt and Vonderau, 2015), this may obscure the fact that data consumption has adverse environmental impacts (Lucivero, 2020). To understand these adverse environmental impacts and how they relate to precision medicine, we start with a review of the broader literature on the environmental (and adverse health) impacts of digital technologies.

## *Carbon emissions*

Heavy carbon dioxide emissions result from the energy required to generate and process large amounts of data. The most recent estimate of the digital sector's contribution to global carbon emissions has been calculated between 2.1% and 3.9% (Freitag et al., 2021). This range reflects some of the uncertainties, controversies and complexities that perplex carbon accounting in the digital sector. This includes the lack of transparency about data centre carbon emissions and the speed of technological innovation which in turn means that calculations may be based on old hardware efficiency figures. It also includes the fact that digital technologies are networks and infrastructures rather than discrete entities, meaning that carbon emissions associated with a particular device or product are difficult to measure (e.g., Horner et al., 2016; Bieser and Hilty, 2018; Koomey and Masanet, 2021). Furthermore, Freitag et al. (2021) and Samuel et al. (2022) both point to how researchers approach carbon accounting differently depending on their discipline, relying on different assumptions and methodologies. Calculating embodied carbon emissions (those emissions associated with the manufacture and transport of digital servers, devices, equipment and servers), while possible (Whitehead and Adrews, 2015), also presents challenges because any emissions attributed to a specific digital material are likely to be entangled with those of other economic sectors (Pierce and Jameton, 2004). This is particularly relevant if we consider the environmental impacts of digital technologies specifically used for precision medicine. This is because precision medicine only uses a small proportion of digital infrastructures, and so it is difficult to dis-entangle exactly what the environmental impact is for this particular field. Nevertheless some data are available for consideration. First, it has been estimated that healthcare data overall make up roughly 6% of all digital data in the datasphere, and this is only likely to increase given that it is the fastest growing sector.[9] As such, considering the environmental impacts of data-driven precision medicine is important. Second, while the environmental impacts of data-driven precision medicine – for example, those related to genomics, and/or the use of natural language processing for analysing electronic medical records – have not yet been studied to any great extent, they are likely to have energy-intensive needs. For example, the energy required to train one particular model in precision medicine research – a deep learning artificial intelligence model (BERT[10] based model without hyperparameter tuning) on a graphics processing unit (GPU) (Rasmy et al., 2021) – has been calculated as equivalent to a trans-American flight (Strubell et al., 2019).[11] Furthermore, a recent study calculated the energy required to conduct a genome-wide association study on biobank data for just one disease trait, to be equivalent to driving about 30 or 100 km, depending on the software used (Grealey et al., 2021).

The digital sector has worked hard recently to drive efficiency gains.[12] Tools available to quantify the carbon footprint of a piece of software are improving (Anthony et al., 2020; Rae et al., 2022) and 'off the shelf' energy-efficient computing hardware and software are also increasingly gaining attention (e.g., Marković et al., 2020).[13] Some scholars predict that likely improvements in energy efficiency and the move to renewable energy will relieve at least some of the above concerns (Malmodin and Lundén, 2018; Giles, 2019), with many hyperscalers[14] already at or heading to net-zero carbon use. However, Blair (2020) argues that the pace of data-driven innovation could outpace the world's renewable energy sources (Blair, 2020). Other scholars stress that it would be remiss to view renewables as a solution to the problem (Morozov, 2013) given that they have their own environmental impacts. For example, with their use (e.g., where they are placed, their effects on the landscape and biodiversity, as well as – for offshore wind – their potential effects on sea temperature),[15] as well as the materials used for their construction (Bihouix, 2020; Mills, 2020) especially rare mineral extraction, which is increasing rapidly to satisfy global demands (Bolger et al., 2021; Voskoboynik and Andreucci, 2021).

Moreover, research has explored the rebound effects of digital technologies, that is, the effects that come from improvements in efficiency. There is now a significant research literature that shows that while increases in energy efficiency may offer environmental advantages in the short term, they will also very likely lead to an increase in consumption in the longer term (Takahashi et al., 2004; Alcott, 2005; Hilty et al., 2006; Börjesson Rivera et al., 2014). We see this in precision medicine, with more health data being collected stored and analysed. In fact, the collection of ever-increasing amounts of (health) data from both clinical and non-clinical (environmental, social media, passive [sleep, heart rate, etc.]) sources allows precision medicine researchers to use ever more powerful (and energy hungry) algorithms to answer endless health-related research questions. One example is digital phenotyping – a precision medicine field developed specifically because the increases in digital efficiency have allowed the collection and analysis of tremendous swaths of data. Digital phenotyping uses machine learning techniques to analyse moment-by-moment individual data from personal sensors and smartphones (social media data, sleep, location, phone records, heart rate, etc.) to improve the diagnoses for targeted intervention (Insel, 2017). Such trends in artificial intelligence growth have led to increasing model size and energy consumption (Wu et al., 2022).

### Impacts of resource extraction

The datafication of health and the move to data-driven precision medicine practices also contribute to the global demand for mineral and metal consumption associated with developing digital infrastructures. Practices associated with mineral extraction often lack regulation, particularly in low-to-middle income countries (LMICs). Mining-associated harms are numerous (Mancini et al., 2021) and include respiratory illness, injuries, cancers and adverse mental health. Community health risks occur through exposure to the air, water, soil and noise pollution that come from mineral extraction and (highly toxic) processing and manufacturing (Harris et al., 2015; Schwartz et al., 2021). A recent global census of 406 lower-to-middle income countries' mining-related hazardous waste sites – affecting an estimated 7.5 million people – revealed that arsenic, lead and mercury, are all strongly associated with adverse health effects, contributing more than three-quarters of the environmental risks at these sites (Caravanos et al., 2013). Responsible mining is now an important issue[16] legislation and ethical codes are enforced in many countries (Arvanitidis et al., 2017; Global Reporting Initiative, 2019; Ayeh and Bleicher, 2021) and have led to several improvements in practice (Deberdt and Billon, 2021). However, poor practices also continue (Bilham, 2021), often attributed to gaps in the regulation (Magallón Elósegui, 2020)[17] or to the fact that initiatives are often developed by powerful companies who shape the discourse and neglect important stages of the mining life cycle (Phadke, 2018), and who outsource responsibility 'at a distance' (Calvão et al., 2021; Deberdt and Billon, 2021), disregard complexity (Ayeh and Bleicher, 2021) and do not engage with the social and cultural context of the industry (Hecht, 2012; Mantz, 2018; Smith, 2022). While health-related and other adverse mining-associated impacts are context specific and will vary depending on the type of mining, the mineral being extracted, as well as the economic, political and cultural context (Bilham, 2021), Samuel and Lucassen (2022) have argued that those working in precision medicine must become more aware of these issues in order to mitigate them as much as possible.

### Electronic waste (e-waste)

The digital technology sector produces a massive amount of e-waste that contains hazardous materials such as lead, cadmium, mercury and nickel, making it a major challenge for disposal, especially when the levels of many of these substances exceed permissible limits (Mmereki et al., 2016; Rautela et al., 2021). This includes the data servers and ICT digital infrastructure that is used in precision medicine. A lack of regulation associated with disposal, recycling and resource recovery (Gabrys, 2012; Mmereki et al., 2016; Lepawsky, 2018; Rautela et al., 2021) means that only about one-fifth of e-wastes are formally collected and recycled globally, with a lack of clarity around what happens to the remainder, but the likelihood is that they are dumped on landfills or traded through illegal markets (Forti et al., 2020). Resource recovery from e-waste landfills is a source of livelihood and business opportunities, but unregulated and informal e-waste recycling methods (e.g., open burning, incineration, acid stripping of metals and acid baths) generate hazardous by-products that have been shown to be present at increased levels in those living around informal e-waste sites, seriously affecting their health (Gabrys, 2012; Dai et al., 2020; Ngo et al., 2021; Singh et al., 2021). Furthermore, Lepawsky (2018) argues that e-waste is more than just end-of-life digital products, but also includes the solid, liquid and gaseous toxic waste that comes from the manufacturing of digital products.

### Other environmental impacts

Less literature has explored the effects of digital technologies on water consumption and biodiversity, though some exist (e.g., Ristic et al., 2015; Mytton, 2021; Lei and Masanet, 2022). Data centres consume water indirectly through electricity generation (often thermoelectric power) and directly through cooling the ICT equipment which generates substantial heat (and subsequent loss through evaporation) of water.[18]

### Precision medicine, data-driven technologies and environmental impacts

While the environmental impacts of data-driven and digital technologies have received substantial attention in the academic, policy and news media arena (Gilmore, 2018; Kuntsman and Rattle, 2019; Schwartz, 2019; Department for Environment, 2021), they have received surprisingly little attention in the health sector, or in precision medicine literature. Rather, the literature has largely focused on promised benefits and increased patient autonomy (Samuel and Farsides, 2017; Birk and Samuel, 2020). One exception is Samuel and Lucassen's (2022) recent mapping of the literature exploring specific environmental impacts of data-driven health research, some of which included research associated with precision medicine. These authors show how most studies have focused on developing software and hardware solutions using green IT, that is, an approach to IT that produces minimal waste during its development and operation and promotes recyclability, with less focus on a consideration of the need to think about changes in data practices (Samuel and Lucassen, 2022). This is not always the case – some scholars have highlighted what researchers and clinicians can do to decrease their environmental impact (Rae et al., 2022). Scott et al. (2012) take a specific focus on e-waste in the health sector, promoting reduce, reuse and recycle mottos. Tongue (2019) calls for more differentiation between useful and redundant data when considering which data should be stored in a healthcare

**Table 1.** Ten rules proposed by Lannelongue et al. (2021) to help make computing for health-related purposes more environmentally sustainable

| | |
|---|---|
| 1 | Calculate the carbon footprint of your work |
| 2 | Include the carbon footprint in your cost–benefit analysis |
| 3 | Keep, repair and reuse devices to minimise electronic waste |
| 4 | Choose your computing facility |
| 5 | Choose your hardware carefully |
| 6 | Increase efficiency of the code |
| 7 | Be a frugal analyst |
| 8 | Releasing a new software? Make its hardware requirements and carbon footprint clear |
| 9 | Be aware of unanticipated consequences of improved software efficiency |
| 10 | Offset your carbon footprint |

system given the environmental impacts associated with exponential increases in data collection and processing, and Chevance et al. (2020) have called for 'digital temperance' rather than 'overconsumption and overpromotion' of data in health systems. These latter authors have described three guiding principles to be incorporated into any health-related data-relevant practices: (1) restraint in production, use and promotion of digital technologies; (2) lifecycles instead of waste (cf. the circular economy); and (3) complex systems approaches through inter-disciplinary collaboration. Lannelongue et al. (2021) have proposed a series of 10 rules for health researchers to make computing more environmentally sustainable, which are listed in Table 1 (Grealey et al., 2021). Such rules are particularly relevant to researchers working in the field of data-driven precision medicine, but as yet there has been little literature focussing on initiatives to mitigate the ever-increasing data consumption by precision medicine researchers.

## Conclusion

As healthcare and health research become increasingly 'datafied', assumptions remain that the use of data is 'free' with few or no consequences to the environment. We have brought attention to the environmental impacts of the data-intensive approaches associated with precision medicine. While such approaches only account for a small proportion of the total adverse environmental (and health) impacts associated with digital technologies more generally, and information is limited on the exact environmental impacts of data-driven precision medicine, it is still important to reflect on this in healthcare and research practices.

Data-driven precision medicine researchers need to consider what data is being collected and analysed and why, what will happen to that data, and what impact it may have on health and the environment (good and bad). As we have shown in this review, while the promise of health benefit is a laudable goal for precision medicine research, adverse health effects can also result from the environmental impacts of precision medicine technologies. Furthermore, it remains true that those most likely to benefit from precision medicine will be those less likely to be harmed by the environmental risks attached to it and vice-versa.

There is a range of ways this imbalance might be re-dressed. Precision medicine researchers can ensure that their data is stored in data centres that are powered by renewable energy, and also adopt best practice in procurement and waste disposal. Progress

can be seen from the fact that many data centres are already using efficient data servers, are actively moving towards net zero, and reducing other environmental impacts. Furthermore, environmentally friendly data storage solutions can be found in long-term data storage, which has longer data accessibility speeds, but significantly lower energy costs. Researchers should consider differentiating their data in terms of storage needs so that data that is not anticipated for short-term use can be stored at lower energy costs. Finally, precision medicine researchers developing their own algorithms must be diligent in their research methods to ensure algorithms are only run once they have been carefully checked and piloted. A range of carbon trackers that allow researchers to estimate the carbon emissions associated with their algorithms can help build awareness around the issues.[19] The field of precision medicine must also think more broadly about how to ensure the adverse and beneficial environmental/health impacts of the field are more evenly distributed. This can involve, for example, developing research questions that have (more) global relevance, and for which any potential beneficial health impacts have been considered early in the research process in terms of their global (and national) affordability and accessibility (Samuel and Richie, 2022). Finally, at a policy level, high-energy-consuming technological solutionism through precision medicine must be explicitly balanced with low-tech (and energy) health solutions, such as those that address the social determinants of health. Social science research has long shown that these social determinants of health play a far greater role in health outcomes than a country's technological clinical capabilities (Institute of Medicine (US) Committee on Assuring the Health of the Public in the 21st Century, 2002). Furthermore, increasing the efficiency of digital solutions should not be viewed as a free pass towards continued consumption, but rather efficiency should be viewed as providing the necessary space between society's consumption and the need not to overshoot planetary boundaries (forthcoming). Overall, there is a range of practices that precision medicine researchers, as well as policymakers, should consider to help balance the benefits and adverse environmental impacts of precision medicine.

**Open peer review.** To view the open peer review materials for this article, please visit http://doi.org/10.1017/pcm.2022.1.

**Data availability statement.** Data availability is not applicable to this article as no new data were created or analysed in this study.

**Author contributions.** G.S. wrote the first draft of the manuscript. A.M.L. reviewed and edited the draft.

**Financial support.** This work was supported by the Wellcome under Grant number: 222180/Z/20/Z and 205339/A/16/Z.

**Competing interests.** The authors declare no competing interests exist.

**Ethics standards.** Not applicable, because this article does not contain any studies with human or animal subjects.

## Notes

1. The authors refer to Great Britain, so we assume they mean England, Wales and Scotland.
2. https://www.who.int/initiatives/cop26-health-programme/country-commitments.
3. https://healthcareclimateaction.org/racetozero. Also see Healthcare without Harm's Reducing Healthcare's Climate Footprint, which contains case studies of various hospital's initiatives to reduce their carbon footprint: chrome-extension://efaidnbmnnnibpcajpcglclefindmkaj; https://noharm-

europe.org/sites/default/files/documents-files/4746/HCWHEurope_Climate_Report_Dec2016.pdf.

4. Also see, for example, https://www.pfizer.com/news/announcements/pfizer-announces-commitment-accelerate-climate-action-and-achieve-net-zero; https://www.roche.com/stories/reducing-our-carbon-footprint.

5. See, for example, https://practicegreenhealth.org/topics/water/water; https://www.cvshealth.com/social-responsibility/corporate-social-responsibility/resource-library/transform-health-2021-water-goal.

6. https://www.sustainabilityexchange.ac.uk/leafanewapproachtoachievinglaboratorysus#:~:text=What%20is%20LEAF%3F,environment%20that%20supports%20research%20quality. Also see https://www.mygreenlab.org/; https://slcan.ca/; https://www.mygreenlab.org/.

7. https://www.ukbiobank.ac.uk/learn-more-about-uk-biobank/news/uk-biobank-creates-cloud-based-health-data-analysis-platform-to-unleash-the-imaginations-of-the-world-s-best-scientific-minds.

8. Potentially much more as further records are digitalised, and the propensity for data grows. For example, a Californian health-based network with more than 9 million members is estimated to have between 26 and 44 petapytes of patient data from electronic health records; cited in: Managing the healthcare information stream, Commvault. 2015. chrome-extension://efaidnbmnnnibpcajpcglclefindmkaj; http://webdocs.commvault.com/assets/managing-the-healthcare-information-stream.pdf.

9. https://www.youtube.com/watch?v=DAR0ATh-TPI.

10. Bidirectional encoder representations from transformers.

11. We assume this refers to a flight travelling between America's East-West coasts.

12. Mainly for business reasons, but more recently to address considerations of the environment. For example, see Samuel et al. (2022).

13. Also see Open Compute Project, https://www.opencompute.org; green Data Center Platform, https://www.greendatacenterplatform.com/.

14. Hyperscaled supply of computing power, cloud computing, networking, and so forth. Examples include Amazon, Facebook and Google.

15. Presented by Professor Nicola Beaumont at UKERC Research Conference, 13–14 June 2022, Manchester, UK.

16. See https://www.responsibleminingfoundation.org/.

17. For example, the EU legislation (in contrast to that of the OECD) only requires downstream obligations for those involved in moving and processing the minerals from the extraction site to their incorporation in the final product so leaves out companies that import already manufactured electronic components.

18. Cooling data centres is expensive, which is why you often see companies building data centres in cooler climates. This reduces costs and decreases water consumption, though it does require data to be transferred longer distances to a user device.

19. It is worth noting that these carbon calculators are problematic for a range of reasons, including the types of data/databases they base their calculations on (forthcoming).

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
