## [Reviewer Report]

*Comments to Author*: The article discuss an often underestimated issue: the environmental cost of dataification in the healthcare system, with a alleged focus on precision medicine. I think the article is of value to promote thought, and maybe action, on thjs subject, although the focus on precision medicine seems a bit diffuse at times. In general terms, my specific comments relate to language revision and to the need to discuss progress made to mitigate the problem, so that a more balanced and comprehensive view on the matter is presented to the reader.

* I suggest changing"Duchenne Muscular Dystrophy" to "Duchenne muscular dystrophy".

* In the following sentence, the information between parentheses appears erased (unreadable). This is repeated later in the text, presumably to hide the authors identities, though a simple copy-paste makes the writing visible. Please make sure it can be read in the final manuscript. "These advances inevitably have a significant environmental footprint, which is often justified using consequential narratives of being necessary to improve healthcare and/or that health has intrinsic value so has a free pass not to consider these issues (Samuel, Hardcastle et al. in press)". Furthermore, I believe that the statement that "it is often justified..." is not actually sufficiently supported by a single reference, which on the other hand corresponds to an article in press. Please support the statement with additional references.

* Please remove the duplicated word in "There is now no doubt that that climate change is attributed to human factor".

* Please revise "with this figure only slightly lower -6%-in Great Britain" from a grammar perspective.

* I suggest rewriting "zika" as "Zika".

* "net zero" is sometimes written as "net zero" and sometimes as "net-zero" through the text. Please uniform.

* Please revise the word "aesthetics". I believe the authors meant "anesthetics".

* I think "emission intensive" should be written "emission-intensive". The same applies for " data intensive".

* Please revise the following sentence: "It means using natural resources that are not harvested faster than they can be regenerated...". I suspect a meaning issue there.

* On page 8, line 1, there seems to be a reference included as superindex, although this is not the reference style in the manuscript.

* Authors state "The environmental impacts of other aspects of precision

medicine, for example proteomics and metabolomics, or the use of natural language

processing for analysing electronic medical records, have not been studied in any depth...". Ok, they have not been studied in depth... but are there any studies available?

* Please revise the writting of "this nay obscure the fact that data consumption is no less environmentally problematic than material goods consumption". "nay". Also, part of the text appears in different colours. Fiurthermore, can the authors provide more references to the statement that data consumption is as complicated from an environmental perspective as material good consumption. I believe this is, at least, debatable (for instance, the authors themselves later state that the carbon emissions linked to the digital sector represent a minor proportion of the whole emissions). Are there studies stating the other way around?

* Please change "genome wide association..." to "genome-wide association...".

* Please change "Cambridge university" to "Cambridge University".

* For an objective treatment of the subject, I believe the authors should more deeply discuss positive trends in the digital sectors, if any, related to more environmental-friendly techs (.e.g, increased efficiency). Recent trends in other technologies (e.g., lighting) have brought around a substancial saving both in energy consumption and heat emissions. Aren't any similar advances in the data-storing field, that can partially mitigate the increase consumption in data services? For instance, advances in data compression and transmission tech? Similarly, are there any ongoing or potentially incoming solutions to the e-waste problem? Recycling is the only foreseen measure?

* Although authors intend to focus on healthcare digital data environmental issues, they actually mostly discuss general environmental negative impacts of big data production, transmisison, and storage. Which is the relative contribution of healthcare data, and, in particular, precision medicine-related data, to the environmental issue of datafication? Are there any studies comparing the impact of healthcare datafication versus, for instance, data production linked to entertainment? Are there any positive environmental impacts of dataification to discuss?

---

## [Reviewer Report]

*Comments to Author*: This review is an important and timely article, it is well referenced and reinforces the recommendation that researchers and practitioners in this area should be aware of the broader impact of their work.

It is not specifically mentioned in the abstract, but the paper includes a useful review of the direct impact on health from the increasingly damaging impact that human society has on the environment. This serves to illustrate the necessity for all of us to find a ‘balance’ between the good that we do and the harm that can be caused in that process. This point is well made in the conclusions of the paper, but it might be useful to highlight the review of the impact on health in the abstract.

In the section on “Reducing carbon emissions and other environmental impacts in healthcare”, the examples included in the text are almost all related to England. It would be potentially more useful to the reader to have some other examples from outside England to illustrate these points.

In the review, consideration is given to impact of computing resources that are applied during the data storage and analysis process; however, no explicit mention is made of the impact of AI machine learning/deep learning. It is covered briefly in some of the papers cited but this review would benefit from a short consideration of the explicit impact of high performance computing (e.g. GPUs) as used for machine learning/deep learning as applied to healthcare data analysis and precision medicine.

The following two papers are possible additional references -

Anthony, L.F.W., Kanding, B. and Selvan, R., 2020. Carbontracker: Tracking and predicting the carbon footprint of training deep learning models. arXiv preprint arXiv:2007.03051.

Wu, Carole-Jean, et al. "Sustainable ai: Environmental implications, challenges and opportunities." Proceedings of Machine Learning and Systems 4 (2022): 795-813.

---

## [Editor Report]

*Comments to Author*: As the reviewers suggests, there are some minor revisions, some additional discussion in a few areas and a couple of possible additional references which would benefit the paper.

---

## [Reviewer Report]

*Comments to Author*: The authors have satisfactorily addressed my previous comments; thank you for takin them into consideration.